# Quality of Sex Life in Intestinal Stoma Patients—A Literature Review

**DOI:** 10.3390/ijerph20032660

**Published:** 2023-02-01

**Authors:** Wiktoria Paszyńska, Katarzyna Zborowska, Mariola Czajkowska, Violetta Skrzypulec-Plinta

**Affiliations:** Department of Women’s Health, Faculty of Health Sciences in Katowice, Medical University of Silesia, 40-055 Katowice, Poland

**Keywords:** quality of sex life, stoma surgery, quality of life, reproductive health

## Abstract

Introduction: Stoma surgery may reduce the quality of life, including sex life. A literature review was undertaken to explore what factors impact on the sexual health and sexuality of people with a stoma. Methodology: A review of the literature was undertaken using the online databases Cochrane, PsychInfo, Embase and Pubmed. The search was limited to articles on colostomates and ileostomates in the English language that were peer-reviewed and written in the past 5 years. Results: Intestinal stoma surgery affects many aspects of a patient’s life, including body image, relationship with a partner and quality of sex life. The introduction of perioperative educational programmes for patients qualified for ostomy surgery and their relatives will provide the necessary support in the face of physical and mental difficulties that may be associated with the procedure.

## 1. Introduction

A successful sex life, next to work, family, leisure and entertainment, is an important factor determining the quality of human life. In the opinion of stoma patients, limitations in sexual life after the creation of a stoma are some of the most important contributors to a lower quality of life. 

Sexual activity is often considered a taboo, due to the belief that sex should be associated with a sense of shame, or is an activity reserved exclusively for procreation. As a result, both patients and healthcare professionals avoid the subject of physicality, which is a serious mistake, as a holistic approach to patient care should also take into account their sexuality [1,2]. Despite the fact that the vast majority of patients report unsatisfactory sex life after stoma surgery, most of the activities of medical personnel are focused mainly on areas related to surgical intervention (recovery, self-care, past pathology), and little time is devoted to the sexuality of people with a stoma [3,4]. 

The emergence of sexual dysfunctions in stoma patients may be associated with the underlying disease, but also with the surgery itself or the form of follow-up treatment, especially in the case of malignancies. Surgical procedures in which an artificial anus is created can significantly affect the sexual activity of people undergoing these surgeries. In women, this is manifested by hyposexuality, vaginal dryness and soreness during intercourse, while in men it is manifested by erectile dysfunction, retrograde ejaculation or lack thereof [5]. 

Sexual dysfunctions that occur after surgery in stoma patients can also be caused by mental factors. They result from the patient’s psychological reactions to the fact of having a stoma, which changes a person’s self-image. After surgery, the anus is located on the anterior surface of the abdominal cavity, an area that is often uncovered and highly visible. This is accompanied by a lack of control over the discharge of gases and stool. These circumstances significantly lower the self-esteem of a stoma patient, which causes fear of not being accepted by their spouse or sexual partner [6,7]. 

## 2. Objective of the Study

The aim of this study was to review the literature on the quality of the sex life of stoma patients. 

## 3. Material and Method

The research question chosen to carry out the systematized review was: How does ostomy affect the sex life? This search focused on adult patients who have a digestive stoma (ileostomy or colostomy). The timeframe for published studies was from 2017 to 2021. The literature search was started in September 2021 and ended in December 2021.

Three researchers independently screened the titles and abstracts of the studies found to identify those that met the inclusion criteria. The articles that were not discarded were then read in full text and assessed for selection. Disagreement over the eligibility of studies was solved through discussion and by a fourth reviewer. To assess the quality of the articles, the scientific levels of evidence designed by the US Agency for Healthcare Research and Quality were used. There are five levels of scientific evidence depending on the type of study, the levels with the highest degree of scientific evidence (such as meta-analyses and systematic reviews) being those in the highest part of the scale, and the lower levels being those with less evidence and therefore less reliability.

A data extraction sheet was developed and the data were extracted by the reviewers. Differences were discussed face to face, and when there was consensus the data were included. The variables that were taken into account were: year of publication, level of evidence, research design, type of ostomy, cause of stoma, measures and aim of the study. Data were synthesized and analyzed by the review authors, and discrepancies were solved by consensus. The results were written as a descriptive narrative synthesis and tables were made to collect the variables taken into account.

The literature search was based on a systematic review of the electronic databases Cochrane, Pubmed, Embase and PsychInfo. The search included studies from anywhere in the world from the last 5 years. Older articles were excluded in order to find the most up-to-date scientific papers. The most relevant 13 were selected, and these are summarised in Table 1. 

### 3.1. Inclusion Criteria

The following keywords were used: “stoma” (OR “ostomy”) AND “sexual health” (OR “sexuality” OR “sexual dysfunction” OR “sex”).

The following inclusion criteria were used:1Articles on the sexual functioning of intestinal stoma patients.2Research reports conducted in adults.3Articles written in English.4Reviewed articles.5Articles published in the last 5 years.6Patients with a colostomy and/or ileostomy.7Patients who have had sexual intercourse.8Patients over 18 years of age.9Quantitative and qualitative studies.

### 3.2. Exclusion Criteria

Articles published in a language other than English and articles that did not meet the criteria (after reviewing the abstract or the text) were rejected. 

After a bibliographic search, 215 articles were found, of which 150 were removed for being duplicates or not meeting the inclusion criteria. The remaining 65 articles were assessed by the authors in a primary review based on the reading of the titles and abstracts, discarding those articles (*n* = 33) that did not correspond to the subject of the review. Of these, 32 articles were read in full text by the authors and 18 were excluded because they did not focus on the objective of the work, were nonspecific or were minimally relevant. A total of 13 articles published between 2017 and 2021were included in the systematic review for this study (Table 1).

Of the studies, three were qualitative and the rest were quantitative (descriptive, case-control, prospective, cross-sectional). According to the US Agency for Healthcare Research and Quality, the level of evidence of the majority of the articles was IIb [8,9,12,13,14,15,16,17,18,19,20], followed by level III [10,11].

## 4. Discussion

All analysed studies showed that undergoing an intestinal stoma surgery has a negative impact on the quality of sex life. The most important issues relating to the impact of a stoma on sex life are described below. Sexual dysfunctions after intestinal fistula surgery, changes in body image, lack of partner acceptance, decreased sexual satisfaction, lack of sexual education and problems related to stoma hygiene were the main reasons for a significant decrease in the quality of sexual life of patients with intestinal stoma.

### 4.1. Stoma and Sexual Dysfunctions

In a study of 920 patients with rectal cancer, 25% of women and 34% of men were sexually active one year after diagnosis (at the time of diagnosis these values were, respectively, 29% and 41%). Of these patients, 51% of women and 62% of men had undergone a stoma procedure. The presence of a stoma additionally contributed to a decrease in sexual activity. Women experienced reduced lubrication and more dyspareunia at 1 year compared with the time of diagnosis. In men, erectile dysfunction increased from 46% to 55% at 1 year [8]. 

A study of 43 men with a stoma due to rectal cancer showed that 76% of respondents experienced deterioration in sexual function (65% reported erectile complaints and 27% reported ejaculation problems). Fourteen patients (38%) did not resume sexual activity after surgery. More advanced age and the presence of a stoma negatively affected the sexual function of the patients [9]. 

The sexual function of 50 patients with a colostomy or ileostomy was examined using the Arizona Sexual Experiences Scale and the Golombok–Rust Inventory of Sexual Satisfaction. The control group consisted of 50 healthy people. The study showed that stoma patients had less intercourse than the control group (68% vs. 30%; *p* = 0.01). It was also found that 52% of stoma patients avoided sexual intercourse. In women with intestinal fistula, the incidence of pain during intercourse was significantly higher than in the control group (92% vs. 52%; *p* = 0.02). It was observed that deterioration in the quality of sex life was correlated with deterioration in physical fitness and in mental functioning [10]. 

In another study, 75 Chinese intestinal stoma patients were examined using the Arizona Sexual Experience Inventory Scale. It was observed that patients most often reported problems related to sexual arousal and the ability to achieve orgasm. Women complained of vaginal dryness, whereas men complained of difficulty in getting an erection. In addition, it was shown that age, gender, relationship with a partner, mode of surgery, type of stoma, occurrence of complications, ability to self-care and support of loved ones were factors that affect the occurrence of sexual dysfunctions [11]. 

### 4.2. Stoma and Body Image

Creation of a stoma causes patients to perceive their own body differently. Body image disturbance often leads to impaired social relationships and occupational functioning. A Body Image Disturbance Questionnaire (BIDQ) was used to assess the perception of body image in 41 stoma patients. Impaired body image was shown to be particularly prominent among younger and overweight patients and those living with a temporary stoma. Males had a higher BIDQ score than females. In addition, it was observed that the negative attitude of patients towards surgery correlated with the occurrence of body image disturbances at a later stage. [12].

Through its negative impact on body image, a stoma lowers a patient’s self-esteem and quality of life. Yilmaz et al. examined 110 intestinal stoma patients. Not only body image but also self-esteem and overall quality of life were taken into account. It was found that 75.7% rated their body image very low. Approximately 77% of the respondents had low self-esteem. A stoma has been shown to adversely affect mental, physical, social, spiritual and sex life [13]. 

On the basis of interviews with 10 stoma patients, it was noted that the presence of an intestinal fistula was a source of shame for them. The presence of a stoma bag, fear of leakage, lack of control over passing gas and stool, and unpleasant smell are all problems that contribute to a critical perception of one’s body. Stoma patients emphasised that it is often the family that exacerbates this condition. After the surgery, loved ones perceived the body of the patient as difficult to look at, strange or distorted. This way of thinking caused a significant reduction in stoma patients’ self-esteem [14]. 

### 4.3. Stoma and Intimate Relationships

Partner support is essential for patients to accept their new body. Stoma patients with strong bonds with their partners returned to sexual activity more easily, led happier intimate lives and demonstrated a positive attitude towards their condition. Many patients also stated that the process of adapting to a stoma contributed to the growth and strengthening of their relationship with their partner. A large proportion of the study participants admitted that they initially did not take care of their stoma on their own. Patients who were incapable of independent stoma care relied heavily on their partners. Some respondents found that they began to see their partner as a friend or carer. Kimura et al. suggested that a partner who assumes the role of a carer in the postoperative period may experience difficulty perceiving themselves as a lover [15]. 

Impairment of the physiological bowel movement process, body image disturbance and problems with sexual functioning after ostomy surgery are key elements disrupting the intimate relationships of stoma patients. It turns out that the risk of divorce in cancer patients (malignant neoplasm is one of the main causes of stoma creation) is 1.77 times that of healthy people. Problems in sex life increase the risk of depression. Better relationships with partners were observed in patients who were able to take care of their stoma on their own. A study involving 390 patients showed that 42.6% had problems with intimate relationships. It was shown that women, more often than men, described their relationships as relationships with a strong sense of intimacy. Also, women were more able to talk openly about problems in their relationships. The ability to self-care and the time since surgery were the factors that mattered most in assessing intimate relationships [16]. 

Problems experienced by spouses/partners of patients have been observed to worsen and become more complex with time. They often experience negative emotions, depression, health problems and conflicts in family and professional life. Studies have shown that 80% of spouses of stoma patients are ashamed to look at their partner’s body [17]. 

### 4.4. Stoma and Sexual Satisfaction and Desire

A study involving 59 patients examined the effect of a stoma on sexual satisfaction. Using the Golombok–Rust Inventory of Sexual Satisfaction (GRISS), it was possible to assess the level of sexual satisfaction, the nature of sexual relationships and the sexual problems of patients living with a stoma. Patients’ negative sexual experiences were found to correlate with low self-esteem. The low self-esteem of patients adversely affected the quality of communication with their partner, frequency of sexual intercourse and sexual satisfaction. There was a positive significant relationship between women’s self-esteem and the frequency, communication and satisfaction sub-dimensions of GRISS (*p* = 0 < 0.05). As for the male patients, there was a positive significant relationship between their self-esteem and the frequency, communication, satisfaction, avoidance, impotence and premature ejaculation sub-dimensions of GRISS (*p* = 0 < 0.05). Respondents who had a significant fear of stoma bag leakage during sexual intercourse exhibited less sexual activity and decreased sexual desire and derived much less pleasure from sexual intercourse [18]. One study found that 3 months after ostomy surgery, patients complained that their sexual satisfaction had decreased, but at 12 months there was improvement in both men and women [19]. In the study by Costa et al. evaluating 43 men with a colostomy, it was observed that as many as 37% did not return to sexual activity. It should be noted that the vast majority of men considered their sex life to be important or very important to them. On this basis, it can be concluded that the inability to return to satisfactory sexual activity can significantly reduce the quality of life. The authors of the Turkish study also observed an adverse effect of a stoma on the sex life of the respondents. Stoma patients exhibited a high level of anxiety regarding engaging in sexual activity, complained about difficulties in obtaining sexual satisfaction and refrained from sexual intercourse [9,10].

Smith et al. discussed the impact of creating a stoma on the quality of sex life of a homosexual person. The study participant underwent an ileostomy including rectum removal. This caused him to no longer be able to have anal intercourse, making him feel as if he had lost part of his sexuality [20]. With more than 45,000 participants, the National Survey of Sexual Attitudes and Lifestyles (NATSAL) is the largest study of sexual behaviour in the world. It shows that the percentage of people engaging in sexual experiments, including anal sex, has been steadily increasing since 1990 among both men and women. The incidence of colorectal cancer increases with age, and therefore the number of stoma patients also increases. NATSAL conducted in 2010 showed that 35% of men and 28% of women aged 16–74 had had anal sex. On this basis, it should be concluded that the problem of the deterioration of the quality of sex life among homosexual patients who have undergone ostomy surgery may be more prevalent than has been recognized [21]. 

### 4.5. Stoma Hygiene and Sexual Life

During intercourse, patients with a stoma most often feel anxiety related to leakage of the ostomy appliance, unpleasant smell and noisy passing of gas. These fears have led some patients to be unable to enjoy sexual intercourse [14,15]. A study conducted by Smith et al. showed that the lack of control over passing gas and stool caused shame and embarrassment in the subjects. It should be noted that the study included only patients who underwent an ileostomy, in which a stoma is constructed in the upper gastrointestinal tract, resulting in a large volume of secreted fluid. This additionally increases the fear of soiling and contributes to an increase in sexual aversion [20]. It has been shown that the public’s view on excretion contributes to shaping negative experiences in people living with an intestinal stoma. As early as in childhood, getting dirty begins to be associated with bad behaviour. The loss of bowel control in stoma patients can induce a feeling of regression in behaviour, which poses a serious threat to self-esteem. Kimura et al. noted that study participants frequently reported anxiety regarding ostomy appliance leakage during sexual intercourse. In many cases, it has been found that these problems can be alleviated through proper education. Detailed information provided by the medical staff can help patients and their partners adapt to the new situation [15]. 

### 4.6. Impact of Education on Patients’ Sex Life

Ostomy surgery is meant to improve the patient’s health. However, the procedure affects all aspects of the patient’s life, including psychosocial functioning, body image and sexual health. Cooperation between the patient and the medical staff is crucial for helping the patient adapt to life with a stoma. Such a partnership helps maintain continuity of care, increases the ability to develop practical stoma care skills, shortens the patient’s hospitalisation and reduces the risk of re-admission. It has been observed that patients who are better educated show greater skills in self-care and stoma management. This positively affects the patient’s quality of life, self-esteem and family relationships [16]. The introduction of perioperative educational programmes for patients qualified for ostomy surgery and their relatives will provide the necessary support in the face of physical and mental difficulties that may be associated with the procedure [10].

Patients often do not realise that helping with sexual problems is the responsibility of the medical team. Stoma patients may find it inappropriate to ask the stoma nurse for advice on an issue that is not directly related to their stoma. People may think of sexuality as something that should be within their own sphere of control and therefore may view any sexual dysfunction as a personal failure for which they feel guilty. Patients often do not know that changes in their sexuality or sexual health are inevitable and therefore do not see a reason to talk about them. Studies also show that patients refrain from expressing concerns regarding their sexuality because they would prefer the specialists to raise this issue.

One way to solve problems regarding patient sexuality is the PLISSIT model (permission, limited information, specific suggestions, and intensive therapy). Using this model allows the medical staff to identify the needs of individual patients and plan appropriate measures [17]. 

## 5. Conclusions

Surgery ending in the creation of a stoma is a problem for both young and old people, many of whom do not even try to have intimate relationships and intercourse after the surgery. A very important issue for enabling a return to sexual activity is the proper education of the patient, both before and after the surgery. Sex education should also include the partner in whom the stoma patient often finds support. Surgery can cause many complications. Depending on the type of stoma, the nerves in the perineal area are sometimes damaged, leading to reduced sexual performance. Sexual dysfunctions during convalescence are also psychologically driven and result from the patient’s anxiety. It is crucial for the patient to talk to the medical staff about the importance of a satisfying sex life. The patient should be reassured that there are many forms of therapy that can improve the quality of sex life.

## Figures and Tables

**Table 1 ijerph-20-02660-t001:** Studies selected for inclusion in the literature review.

Author, Country, Year	Objective of the Study	Number of Persons Examined	Stoma Type	Main Findings
Sörensson et al., Denmark, Sweden, 2019 [8].	Assessment of patients’ sexual functioning 1 year after rectal cancer diagnosis.	1085 patients at the time of diagnosis of rectal cancer and 920 patients within 1 year of diagnosis.	Colostomy	The presence of a stoma contributed to a decrease in sexual activity after surgery. Erectile dysfunction is common in men, while vaginal dryness is common in women.
Costa et al., Portugal, 2018 [9].	Evaluation and characterisation of sexual dysfunctions in patients with a stoma due to rectal cancer.	43 patients	Colostomy	The study showed a clear negative impact of a stoma on the quality of sex life of patients.
Bahayi et al., Turkey, 2018 [10].	Assessment of the quality of life and sexual dysfunctions in stoma patients.	50 stoma patients—study group 50 healthy people—control group	Colostomy and ileostomy	It was found that patients with an ileostomy or colostomy had a higher rate of anxiety symptoms and less sexual pleasure and often avoided sexual intercourse.
Zhu et al., China, 2017 [11].	Assessment of sexual functioning of Chinese stoma patients.	75 patients	Colostomy and ileostomy	The results of this study indicate that patients living with a stoma experience sexual dysfunctions. Medical personnel should provide sexual health education for both the patients and their partners.
Jayarajah et al., Sri Lanka, 2017 [12].	The purpose of the study was to evaluate the acceptance of an altered body image following ostomy surgery.	41 patients	Colostomy and ileostomy	Low body image acceptance was associated with having a temporary intestinal fistula, obesity and younger age.
Yilmaz et al., Turkey, 2017 [13].	Assessment of the impact of a stoma on sexual functioning and quality of life.	57 patients	Colostomy	All patients experienced sexual dysfunctions. Having a stoma negatively affected self-esteem and body image, which resulted in a reduction in the quality of life.
Marques et al., Brazil, 2018 [14].	Evaluation of the acceptance of an altered body image of stoma patients.	10 patients	Colostomy and ileostomy	Patients were ashamed of their bodies and believed that having a stoma makes it difficult to perform everyday activities and negatively affects relationships with other people.
Kimura et al., Brazil, 2017 [15].	Evaluation of the quality of sex life of men after the creation of an intestinal stoma due to colorectal cancer.	56 patients	Colostomy	Men most often reported changes in body image and low self-esteem, which negatively affected intimate relationships. Perioperative education significantly helps patients to adapt to the new reality, which results in an improvement in the quality of their sex life.
Du et al., China, 2021 [16].	Assessment of the relationship between the patient’s level of adaptation to life with a stoma and the quality of their intimate relationships.	390 patients	Colostomy	Patients should be educated in order to improve their self-care abilities. Patients who showed greater independence in managing their stoma had a higher quality of sex life.
Türkmenoglu et al., Turkey, 2019 [17].	Identification of problems encountered by the spouses of people with an intestinal stoma and investigation of practices used to solve these problems.	80 patients with partners	Colostomy and ileostomy	Ostomy surgery can have a negative impact on both the patient and their partner, causing difficulties with everyday life activities and psychological, social and economic problems.
Gozuyesil et al., Turkey, 2017 [18].	Evaluation of self-esteem and sexual satisfaction of intestinal stoma patients.	59 patients	Colostomy and ileostomy	Having a stoma negatively affected the patients’ self-esteem. Both women and men reported the occurrence of sexual problems. Sexual dysfunctions caused reduced sexual satisfaction.
Souza et al., Brazil, 2018 [19].	Assessment of the quality of life, including sex life, of patients with an intestinal stoma.	29 patients	Colostomy	It was found that 3 months after the ostomy surgery, patients complained that their level of sexual satisfaction had decreased, but at 12 months there was an improvement.
Smith et al., Great Britain, 2017 [20].	Assessment of the impact of a stoma on body image, relationships with loved ones, self-esteem and sexual functioning.	21 patients	Ileostomy	It was found that an ileostomy can destabilise self-esteem, distort body image and adversely affect sexual functioning.

## Data Availability

MDPI Research Data Policies at https://www.mdpi.com/ethics.

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
