# Peer review of "Quality of Sex Life in Intestinal Stoma Patients—A Literature Review"

_ijerph, 2023, doi:10.3390/ijerph20032660_

Round 1

Reviewer 1 Report

The Authors have undertaken a subject of sexual health and sexuality of stomated patients using the reliable  review of the most relevant to the eight inclusion criteria  fifteen articles  devoted to articles on colostomates and ileostomates published within  the recent 5 years.   It is well know that stoma, through its negative impact on body image, lowers a patient’s self-esteem and quality of life, yet this is still a taboo issue even in the medical milieu. In this context, the effort made by the authors should be highly appreciated  because their paper may increase an awareness of this commonly neglected aspect of the stomated patients  life. The main conclusion that the proper education of the patients, both before and after the surgery, including their partners, is  fundamental is of great importance and should be widely distributed among the professionals dealing with patients with a stoma.

The methodology of the evaluated  review seems adequate for such publications, and my one minor comment concerns  a need of enriching the Introduction with the more recent references - the  ones used date back to a period of 2012-2014.  

Author Response

Good morning,

I am enclosing the revised manuscript and cover letter for corrections in line with the reviewers' comments.

Referring to the comments in the first review, the introduction has been enriched with newer scientific reports.
(3. Sutsunbuloglu E, Vural F. Evaluation of Sexual Satisfaction and Function in Patients Following Stoma Surgery: A Descriptive Study. Sexuality and Disability. 2018; 36: 349-361. 4. García-Rodríguez MT, Barreiro-Trillo A, Seijo-Bestilleiro R, González-Martin C. Sexual Dysfunction in Ostomized Patients: A Systematized Review. Healthcare (Basel). 2021; 9(5): 520.                                                                                                                                                
2.     Referring to the comments in the second review:
1)     In the literature, item 21(earlier item 20) is presented in a clearer way.
2)     A study conducted in 2016 was removed from the review.
3)     At the beginning of the discussion, it was explained why certain aspects were analysed.
4)     The methodology was written following the Preferred Reporting Items for Systematic Reviews and Meta Analyses (PRISMA) protocol.
5)     The results of the study indicate the number of articles obtained from the review of individual databases (total number of articles, how many met the criteria, how many were excluded, how many were analyzed).
6)     Efforts were made to differentiate the analyzes according to the sex of the respondents. However, it was not easy, because not all study authors used the division of the studied factors depending on gender.
In the stoma and sexual dysfunction analysis, a gender breakdown was made in the study Sörensson  M et al.
In the stoma and body image analysis, a gender breakdown was made in the study Jayarajah et al.
In the stoma and intimate relationships analysis, a gender breakdown was made in the study Du et al.
In the Stoma and sexual satisfaction and desire analysis, a gender breakdown was made in the study Gozuyesil et al.

Reviewer 2 Report

The article is interesting, but needs improvement. It is necessary to refine the methodology according to the PRISMA methodology [BMJ 2021;372:n71].

The time period should definitely be improved - the authors analyzed the works from 2016.

They should include a figure from the review of each databases { the total number of articles, how many met the criteria, how many were excluded, how much has been analyzed, the type of article (orginal, review, series report?}} . 

In my opinion the Authors should try to differentiate the analyzes according to the gender of the respondents.

In the references item 20 needs suplemented.

In the discussion the Authors should explain why such aspects are analysed. 

Author Response

(The authors gave the same response as above.)

Round 2

Reviewer 2 Report

thank you for the improvement according earlier suggestions